# The estimated hepatitis C seroprevalence and key population sizes in San Diego in 2018

Adriane Wynn[1]*, Samantha Tweeten[2], Eric McDonald[2], Wilma Wooten[2], Kimberley Lucas[3], Cassandra L. Cyr[1], Maricris Hernandez[1], Franchesca Ramirez[4], Corey VanWormer[5], Scott Suckow[6], Christian B. Ramers[7], Natasha K. Martin[1]

**1** Division of Infectious Diseases and Global Public Health, University of California San Diego, San Diego, CA, United States of America, **2** San Diego County Health and Human Services Agency, San Diego, CA, United States of America, **3** California Correctional Health Care Services, San Diego, CA, United States of America, **4** Rady Children's Hospital San Diego, San Diego, CA, United States of America, **5** San Diego Blood Bank, San Diego, CA, United States of America, **6** American Liver Foundation Pacific Coast Division, **7** Laura Rodriguez Research Institute, Family Health Centers of San Diego, San Diego, CA, United States of America

* awynn@health.ucsd.edu

**Data Availability Statement:** All relevant data are within the paper and its S1 Fig, S1 Table and S1 Text.

**Funding:** Natasha K Martin acknowledges funding from the National Institute for Allergy and

## Abstract

### Background

The Eliminate Hepatitis C San Diego County Initiative was established to provide a roadmap to reduce new HCV infections by 80% and HCV-related deaths by 65% by 2030. An estimate of the burden of HCV infections in San Diego County is necessary to inform planning and evaluation efforts. Our analysis was designed to estimate the HCV burden in San Diego County in 2018.

### Methods

We synthesized data from the American Community Survey, Centers for Disease Control and Prevention, California Department of Public Health, Public Health Branch of California Correctional Health Care Services, San Diego Blood Bank, and published literature. Burden estimates were stratified by subgroup (people who inject drugs in the community [PWID], men who have sex with men in the community [MSM], general population in the community [stratified by age and sex], and incarcerated individuals). To account for parameter uncertainty, 100,000 parameter sets were sampled from each parameter's uncertainty distribution, and used to calculate the mean and 95% confidence interval estimates of the number of HCV seropositive adults in San Diego in 2018.

### Findings

We found there were 55,354 (95% CI: 25,411–93,329) adults with a history of HCV infection in San Diego County in 2018, corresponding to an HCV seroprevalence of 2.1% (95% CI: 1.1–3.4%). Over 40% of HCV infections were among the general population aged 55–74 and one-third were among PWID.

Infectious Diseases and National Institute for Drug Abuse (R01 AI147490), and the University of San Diego Center for AIDS Research (CFAR), a NIH funded program (P30 AI036214). Adriane Wynn acknowledges funding from the National Institute on Alcohol Abuse and Alcoholism (K01 AA027733) and the National Institute on Drug Abuse (T32 DA023356).The funders of the study had no role in study design, data collection, data analysis, data interpretation, or writing of the report. The corresponding author had full access to all the data in the study and had final responsibility for the decision to submit for publication.

**Competing interests:** I have read the journal's policy and the authors of this manuscript have the following competing interests: Natasha K Martin has received unrestricted research grants unrelated to this work from Gilead and Merck.

## Conclusion

Our study found that the largest share of infections was among adults aged 55–74, indicating the importance of surveillance, prevention, testing, and linkages to care in this group to reduce mortality. Further, programs prioritizing PWID for increased HCV testing and linkage to care are important for reducing new HCV infections.

## Introduction

The hepatitis C virus (HCV) causes a chronic liver infection that can result in significant liver damage, disability, cancer, and death. More than 41,000 Americans were estimated to be newly infected with HCV in 2016 [1] and HCV has been noted to kill more Americans than any other nationally notifiable infectious disease, prior to COVID-19 [2]. HCV can be easily identified with available blood tests, and 8–12 week oral treatments can cure nearly all infected patients with few side effects [3]. Unfortunately, most individuals with HCV are not aware that they are infected or are not being treated [4].

The World Health Organization (WHO) and U.S. National Academies of Sciences, Engineering, and Medicine have set HCV elimination strategies with the goals of reducing new HCV infections by 80% and HCV-related deaths by 65% by 2030 [4, 5]. In response, state and local officials, including those in San Diego County, have initiated their own HCV elimination efforts. The Eliminate Hepatitis C San Diego County Initiative was approved by the San Diego County Board of Supervisors in January 2020 and the recommendations report was approved on March 10, 2020. The initiative is composed of a public-private partnership that seeks to make recommendations and establish a roadmap on how to achieve the WHO HCV elimination targets through improved screening strategies and linkage to care and treatment, addressing and removing barriers to cure, reducing harm and preventing reinfection, and supporting policies that facilitate HCV elimination.

An estimate of the burden of HCV infections in San Diego County is necessary to inform elimination planning efforts and to provide a foundation to assess HCV elimination resource needs. Currently, San Diego County conducts core HCV surveillance through mandated reporting of positive HCV antibody and positive RNA test results. However, reliance on this reporting alone does not provide a valid measure of prevalence as it excludes the undiagnosed and fails to account for those who have died, moved away, or were cured. Our study addresses this gap in knowledge by estimating the burden of HCV among adults in San Diego County in 2018.

## Methods

### Overall approach

The burden of HCV was estimated among adults in San Diego County through synthesizing available published and unpublished data on populations at risk and HCV seroprevalence (anti-HCV positivity, a marker of past or current infection) in each group, obtained through literature reviews and data requests to public agencies. The mutually exclusive groups examined were people who inject drugs (PWID) in the community, defined as those who have injected in the past 12 months; men who have sex with men (MSM) in the community; the general population in the community, excluding the aforementioned groups (general population stratified by age and sex); and people who are incarcerated in California with San Diego as their county of commitment.

First, we estimated the population sizes of the above-mentioned groups using data from the American Community Survey (ACS), San Diego National HIV Behavioral Survey (NHBS), the Public Health Branch of California Correctional Health Care Services, and published literature. Next, we estimated HCV seroprevalence for each group using data from the San Diego Blood Bank, the Study of Tuberculosis, AIDS, and Hepatitis C Risk (STAHR II), the NHBS, the Public Health Branch of California Correctional Health Care Services, and published literature. Finally, we multiplied the prevalence by the population size for each group to obtain estimates of the number of individuals with current or past HCV in San Diego County.

## Population size estimates

**General population.** To estimate the overall adult population size of San Diego County, the adult (aged 18+) general population size estimate for San Diego County was obtained from the ACS 2018 (the most recent survey year available) [6]. The ACS provides population estimates grouped by gender and 5-year age groups. Point estimates and 95% confidence intervals by gender were produced in the following age groups: 18–54, 55–74, and 75+ years. These groups were chosen to address the increased risk of HCV among those known as the "1945–1965 birth cohort" of baby boomers (who were aged 53–72 years in 2018) [7]. According to the Centers for Disease Control and Prevention (CDC), the higher HCV seroprevalence among adults born from 1945–1965 corresponds to the high number of incident infections that occurred among young adults in the 1970s and 1980s due to the high frequency of injecting drug use during that period or receipt of contaminated blood transfusions [8]. From this general population estimate, we subtracted the number of individuals incarcerated in the sole state prison located in San Diego County, as these individuals were included in the ACS estimates, but may not have been San Diego County residents prior to incarceration. Healthcare for people in state prisons is handled at the state level, and individuals released to other counties after incarceration in the state prison located in San Diego County are not deemed San Diego County residents for public health purposes. However, individuals in state prisons across California who were originally committed from San Diego County were included separately (see 'People Incarcerated in California State Prisons'). The final community general population estimates were then obtained by subtracting estimates for the PWID and MSM risk groups described below.

**People Who Inject Drugs (PWID).** We define the PWID population as individuals with recent (past year) injecting drug use; therefore, individuals with past injecting drug use risk would be classified as part of the general population in our estimates). Estimates of the population size of PWID were derived from a study by Tempalski et al. [9], which estimated the annual population sizes of current (past year) injectors, from 2002 to 2007; more recent size estimates for PWID were unavailable for San Diego. We used the Tempalski point (24,991), minimum (3,751), and maximum (49,503) estimates, which were derived from HIV testing and counseling and drug treatment services use data, the rate of incident AIDS diagnoses among PWID, and previously published estimates for San Diego. Based on a population size of 2.6 million adults in San Diego County in 2018, the mean estimate corresponds to a PWID prevalence of 0.74%. We allocated the number of PWID by gender according to the Tempalski estimates, and by age using the age distributions among PWID, in the 2018 San Diego National HIV Behavioral Surveillance [10]. We also conducted a sensitivity analysis by multiplying Tempalski's 2007 San Diego PWID prevalence estimate (1.24%) with the 2018 San Diego adult population.

**Men Who Have Sex with Men (MSM).** Estimates of MSM were based on a recent study by Grey et al., which used data from the ACS (2009–2013) to calculate the proportion of men

who had sex with men within the past five years in each U.S. county [11]. First, the MSM prevalence among men in San Diego was obtained from Grey (6.7% and sampled from a beta distribution). Next, this prevalence was multiplied by the numbers of men by age group in San Diego County from the ACS (sampled from a uniform distribution) to obtain the total number of MSM in San Diego County.

While there is some overlap between the PWID and MSM, other studies in similar U.S. settings suggest that MSM who inject drugs and PWID who are men who have sex with men are generally distinct and should be grouped according to their "primary characteristic." An analysis of the San Francisco National HIV Behavioral Surveillance MSM and IDU cycle data found different HCV prevalence estimates between MSM included in the PWID survey and PWID included in the MSM survey [12]. This analysis also found different patterns in drug use and levels of education and employment between the two groups. As such, it is assumed these groups are mutually exclusive with regard to their HCV risk.

**People incarcerated in California State Prisons.** The total number of individuals with San Diego as their county of commitment, who were incarcerated in all California state prisons on December 31, 2018, were obtained from the Public Health Branch of California Correctional Health Care Services. Data were not available on individuals incarcerated in prisons outside of California who had San Diego as their county of commitment, nor on San Diego residents detained in federal prisons or detention facilities.

### HCV seroprevalence estimates

**Literature review.** PubMed, Embase, and Web of Science were searched between April and May 2019, for HCV prevalence or incidence estimates in San Diego with no language restriction (see data in S1 Text and S1 Fig). After duplicates were removed, the combined search yielded 738 results, from which studies older than 20 years (published prior to 1999) were removed, leaving 703 references for title review. All title/abstracts were double screened, with conflicts resolved by a third reviewer. Abstracts were excluded if they were unrelated to HCV epidemiology, reported studies outside of the U.S., or dealt only with treatment outcomes. Following title/abstract review, 60 references were retained for full article review. From this review, seroprevalence estimates were obtained for PWID and HIV-infected MSM described below; no general population estimates were identified. HCV incidence estimates were only available for HIV-infected MSM [13], and were not used in this study as our primary aim was to estimate burden.

**General population.** To inform the seroprevalence estimates for the general population, data were used for first-time allogeneic blood donors at the San Diego Blood Bank, across a ten-year period (2009–2019), who were San Diego residents at the time of donation (n = 151,684). Donors' ages were normalized to age in 2018 and combined into the following age groups: 18–54, 55–74, and 75+ years to account for the higher seroprevalence among the birth cohort [7]. To account for potential bias from the 'healthy donor effect' [14] and selection bias related to blood donor eligibility criteria (which exclude PWID, MSM, and others), a weighted adjustment was applied to prevalence as an 'inflation factor' of 4.9 (95% CI: 2.2–7.7, sampled from a uniform distribution), consistent with a previous analysis [12]. As chronic prevalence estimates were unavailable, we were not able to include these in our analysis.

**PWID.** For this analysis, seroprevalence estimates were used from the STAHR II, a longitudinal cohort study that recruited 574 PWID across San Diego County by direct street and venue-based outreach and targeted advertising between 2012 and 2014 [15]. Among STAHR II participants, 66% (95% CI: 61–70%) were HCV seropositive [15].

**MSM.** A study was identified documenting HCV prevalence among HIV-infected MSM in San Diego, but no data were available among HIV-negative MSM [16]. Studies from other

settings indicate that HCV prevalence among HIV-negative MSM is similar to the general population; consistent with a study from 1999–2003 in San Diego among MSM who do not inject drugs [16]. As such, an estimate of the number of MSM with HIV was first generated by multiplying the aforementioned MSM population size estimate by the weighted HIV prevalence observed among MSM from the 2017 San Diego NHBS MSM survey data (20%; 95% CI: 11.1–28.9%, sampled from a beta distribution). Among HIV-negative MSM, the general population HCV prevalence among men by age from the ACS data were applied. Among HIV-positive MSM, an HCV seroprevalence estimate of 16.5% (95% CI: 15.5–17.6%, sampled from a beta distribution) was used from a study reporting 2008–2012 data [17].

**People incarcerated in state prisons.** The number of all individuals with San Diego as their county of commitment who were incarcerated in California prisons, on December 31, 2018, and diagnosed with HCV-antibody were directly obtained from the Public Health Branch of California Correctional Health Care Services. California Correctional Health Care Services implemented routine HCV screening, in 2016, and since the introduction uptake rates were extremely high (>90%), indicating that the vast majority of cases in prison have been identified.

### Uncertainty and sensitivity analysis

To account for parameter uncertainty, 100,000 parameter sets were sampled from each parameter's uncertainty distribution, and used to calculate the mean and 95% confidence interval estimates of the number of HCV seropositive adults in San Diego in 2018.

We additionally performed a sensitivity analysis using the 2007 PWID prevalence rates applied to the 2018 adult population size [18]. All calculations were performed in MATLAB 2019.

## Results

Based on this available epidemiological data, it was estimated that in 2018 there were approximately 55,354 (95% CI: 25,411–93,329) adults aged over 18 with a history of HCV infection (HCV seropositive) in San Diego County. Our estimate includes 17,005 (95% CI: 5,780–28,840) PWID; 4,086 (95% CI: 2,004–6,940) MSM; 2,018 (95% CI: 1,950–2,086) people incarcerated in California state prisons with San Diego as their county of commitment; 22,066 (95% CI: 11,232–37,327) in the 1945–1965 birth cohort, and 10,179 (95% CI: 4,377–18,204) in the general population (Table 1). This corresponds to a total estimated HCV seroprevalence of 2.1% (95% CI: 1.1–3.4%) in San Diego.

The distribution of infections among risk groups is shown in Table 2. Among all infections, more than 40% were among persons aged 55–74 in the general population in 2018 (this age grouping was the closest to the 1945–1965 birth cohort [aged 53–72 in 2018] permitted with grouped ACS data). Nearly one-fifth of infections were among the general population outside of these age groups. Additionally, although PWID make up only 1% of the San Diego County population, we estimate one-third of HCV infections in the County were among PWID.

A sensitivity analysis applying the 2007 PWID prevalence rate of 1.24% (minimum: 0.19%, maximum: 2.46%)(9) to the 2018 San Diego County adult population, resulted in a larger number of PWID, at 33,915 (95% CI: 11,344–57,528), compared to 25,935 (95% CI: 8,976–43,678) for the main analysis (S1 Table). The resulting estimation of PWID with a history of HCV infection was 22,270 (94% CI: 7,412–37,906), which was 30% higher than the main analysis, with a total estimated 17,005 (95% CI: 5,780–28,840) adults with a history of HCV infection. In this sensitivity analysis, PWID made up a larger percentage of all San Diego HCV infections (37% compared to 31% in the main analysis).

**Table 1. Estimated population size and HCV seroprevalence, San Diego County, 2018.**

| Subpopulation | | Population Size Point Estimate | 95% Confidence Interval | | HCV seroprevalence (anti-HCV) Point Estimate | 95% Confidence Interval | | # HCV seropositive | 95% Confidence Interval | |
|---|---|---|---|---|---|---|---|---|---|---|
| PWID | | 25,935 | 8,976 | 43,678 | 0.6560 | 0.6140 | 0.6942 | 17,005 | 5,780 | 28,840 |
| MSM | | 88,763 | 61,559 | 120,549 | | | | 4,086 | 2,004 | 6,940 |
| | HIV positive | 17,038 | 9,063 | 27,565 | 0.1654 | 0.1551 | 0.1759 | 2,818 | 1,494 | 4,578 |
| | HIV negative (18–54) | 49,767 | 36,552 | 64,273 | 0.0072 | 0.0032 | 0.0118 | 359 | 149 | 649 |
| | HIV negative (55–74) | 17,479 | 12,722 | 22,811 | 0.0470 | 0.0242 | 0.0808 | 846 | 352 | 1,527 |
| | HIV negative (75+) | 4,479 | 3,222 | 5,900 | 0.0128 | 0.0018 | 0.0382 | 63 | 9 | 186 |
| General Population (excluding other groups)[bcd] | | | | | | | | | | |
| | Men 18–54 | 833,594 | 792,414 | 869,444 | 0.0072 | 0.0032 | 0.0118 | 6,053 | 2,696 | 9,814 |
| | Men 55–74 | 290,355 | 269,344 | 311,565 | 0.0470 | 0.0242 | 0.0808 | 13,769 | 7,225 | 22,395 |
| | Men 75+ | 76,566 | 69,489 | 83,691 | 0.0128 | 0.0018 | 0.0382 | 985 | 148 | 2,856 |
| | Women 18–54 | 838,611 | 817,451 | 860,791 | 0.0037 | 0.0018 | 0.0065 | 3,141 | 1,533 | 5,534 |
| | Women 55–74 | 345,081 | 324,071 | 366,751 | 0.0240 | 0.0122 | 0.0427 | 8,297 | 4,007 | 14,932 |
| | Women 75+ | 115,301 | 106,901 | 124,111 | 0.0000 | 0.0000 | 0.0000 | 0 | 0 | 0 |
| | Total general pop | 2,499,508 | 2,379,670 | 2,616,353 | | | | 32,245 | 15,609 | 55,531 |
| People incarcerated in California state prisons[a] | | 8793 | | | 0.2295 | 0.2218 | 0.2372 | 2,018 | 1,950 | 2,086 |
| TOTAL | | | | | | | | 55,354 | 25,411 | 93,329 |

Notes: [a]Individuals incarcerated in 12/31/18 in California with San Diego as their county of commitment.

[b]Excluding other risk populations above.

[c]Blood donor data adjusted by an inflation factor of 4.9 (CI 2.2–7.7) for 'healthy donor effect' as per Facente et al. 2018.

[d]Closest age groups to the aged 55–74 1945–1965 in 2018 based on ACS age groupings.

## Discussion

It was found that approximately 55,354 (95% CI: 25,411–93,329) adults had a history of HCV infection in San Diego County in 2018 corresponding to a seroprevalence of 2.1% (1.1–3.4%).

**Table 2. Summary of estimated HCV burden by subpopulation in San Diego County, 2018.**

| Subpopulation | # HCV seropositive | | | | |
|---|---|---|---|---|---|
| | Point estimate | 95% confidence interval | | % of all SD HCV seropositives | % of Subpopulation in SD population |
| PWID | 17,005 | 5,780 | 28,840 | 31% | 1% |
| MSM | 4,086 | 2,004 | 6,940 | 7% | 3% |
| People incarcerated in California state prisons | 2,018 | 1,950 | 2,086 | 4% | 0.3% |
| 1945–1965 birth cohort[a] | 22,066 | 11,232 | 37,327 | 40% | 24% |
| General Population | 10,179 | 4,377 | 18,204 | 18% | 71% |

Note: [a]Due to American Community Survey data age grouping, the 1945–1965 birth cohort includes those aged 55–74 years.

This compares with national estimate of a 1.7% HCV seroprevalence among adults [18]. Our point estimate is higher than the California Department of Health Office of Viral Hepatitis Prevention's estimate that there were 37,000 County residents living with a known past or current diagnosis of HCV in 2017, which is based on a de-duplicated database of all people testing positive for HCV in San Diego. This difference suggests that approximately 67% of County residents with a past or current HCV infection have been diagnosed, which is higher than a national estimate that half of all those with a chronic HCV infected were diagnosed and aware, yet still highlights that 33% are undiagnosed [19].

Our study found that the largest share of infections was among adults in the 1945–1965 birth cohort, followed by PWID. Importantly, our study also found that nearly one-fifth of infections were among the general population outside of the 1945–1965 birth cohort, indicating the importance of surveillance, prevention, testing, and linkages to care in this group as well. In 2020, the CDC expanded HCV screening guidelines to recommend a one-time screen for all adults aged 18 years and older, in addition to risk-based screening [20]. Although the COVID-19 pandemic may have interrupted HCV screening efforts [21], universal screening of adults in San Diego could enable the detection of a substantial fraction of HCV infections. Following diagnosis, short-duration direct-acting antivirals (DAAs) are highly effective at curing >90% of individuals,(3), preventing HCV-related mortality and could also prevent transmission [22].

Achieving the twin HCV elimination goals of reducing HCV mortality and HCV incidence by 2030 may require prioritizing particular subpopulations for treatment and prevention interventions. Meeting the mortality goal requires treating individuals with more advanced liver disease; these individuals may be older without ongoing transmission risk, such as those in the 1945–1965 birth cohort. Whereas, reducing incidence requires treatment and prevention interventions for those with ongoing risk such as PWID; these individuals may be younger with less advanced disease. Largely due to the expanding opioid epidemic [2], acute HCV diagnoses are increasing in the U.S., particularly among younger PWID who are between 18 and 35 years old [3, 4]. The risks associated with injecting drug use are estimated to account for 77% of ongoing HCV transmission in North America [23]. Harm reduction interventions such as opiate substitution therapy (OST) and needle/syringe programs (NSPs) have been found to be both effective and cost-effective to prevent HCV acquisition [24]. Thus, scale-up of combination harm reduction and treatment services are required to achieve incidence reduction goals [22, 25].

This study has a number of limitations. First, there is considerable uncertainty in our burden estimation, driven largely by uncertainty in PWID population size estimates which are dated and merit updating. Second, we were unable to determine viremic chronic infection burden as RNA data were not available from blood donors in San Diego County. An estimated one-quarter of individuals spontaneously clear infection and do not progress to chronic infection, with slightly lower clearance rates among individuals with HIV [26, 27]. Information on chronic (active) infection is essential for determining HCV treatment need in San Diego County and for monitoring progress towards elimination as individuals are cured. Information on chronic (active) infection is therefore essential for determining HCV treatment need in San Diego County and for monitoring progress towards elimination as individuals are cured. Thus, improved surveillance systems for chronic HCV and HCV-related mortality, including reporting of negative RNA results to identify virologic cures, will allow for generation of estimates of chronic infection burden and provide more robust evidence of progress towards elimination in the future.

The burden of HCV in San Diego County was estimated, which will inform policy-makers on the levels of resource allocation necessary and possible sub-populations for prioritization.

However, in order to determine the level and combination of interventions (e.g., harm reduction and HCV treatment) required to achieve the HCV elimination targets, more research and a more robust public health surveillance infrastructure, including epidemic modeling, is required.

## Supporting information

**S1 Text. Literature review search strategy.**
(DOCX)

**S1 Fig. Flow diagram for the review.**
(DOCX)

**S1 Table. Results from a sensitivity analysis applying the 2007 PWID prevalence rate of 1.24% (min: 0.19%, max: 2.46%) to the 2018 San Diego County adult population.**
(DOCX)

## Acknowledgments

We thank Rachel McLean and Lauren Stockman from the California Department of Public Health and Adam Bente (Family Health Centers of San Diego); Onika Chambers, Anna Flynn, Marisa Ramos (California Department of Public Health Office of AIDS) from the National HIV Behavioral Surveillance San Diego for their assistance with obtaining and analyzing NHBS and California Correctional Health Care Services data as well as providing feedback on this manuscript.

## Author Contributions

**Conceptualization:** Adriane Wynn, Scott Suckow, Natasha K. Martin.

**Data curation:** Adriane Wynn, Samantha Tweeten, Eric McDonald, Wilma Wooten, Kimberley Lucas, Cassandra L. Cyr, Maricris Hernandez, Franchesca Ramirez, Corey VanWormer, Scott Suckow, Christian B. Ramers, Natasha K. Martin.

**Formal analysis:** Adriane Wynn, Cassandra L. Cyr, Maricris Hernandez, Franchesca Ramirez, Natasha K. Martin.

**Funding acquisition:** Adriane Wynn, Natasha K. Martin.

**Investigation:** Cassandra L. Cyr, Maricris Hernandez, Franchesca Ramirez.

**Validation:** Samantha Tweeten, Eric McDonald, Wilma Wooten, Kimberley Lucas, Corey VanWormer, Scott Suckow, Christian B. Ramers.

**Writing – original draft:** Adriane Wynn, Natasha K. Martin.

**Writing – review & editing:** Adriane Wynn, Samantha Tweeten, Eric McDonald, Wilma Wooten, Kimberley Lucas, Cassandra L. Cyr, Maricris Hernandez, Franchesca Ramirez, Corey VanWormer, Scott Suckow, Christian B. Ramers, Natasha K. Martin.

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
