## [Decision Letter · Decision Letter 0]

10 Feb 2021

PONE-D-20-38492

Estimated hepatitis C prevalence and key population sizes in San Diego

PLOS ONE

Dear Dr. Wynn,

Thank you for submitting your manuscript to PLOS ONE. After careful consideration, we feel that it has merit but does not fully meet PLOS ONE’s publication criteria as it currently stands. Therefore, we invite you to submit a revised version of the manuscript that addresses the points raised during the review process.

Your manuscript was reviewed by 2 experts in the field. Both identified several extremely critical issues in your submission and produced strong recommendations. It is important that you carefully consider all comments and provide detailed point-by-point responses.

We look forward to receiving your revised manuscript.

Kind regards,

Yury E Khudyakov, PhD

Academic Editor

PLOS ONE

Journal Requirements:

"Natasha K Martin acknowledges funding from the National Institute for Allergy and Infectious Diseases

and National Institute for Drug Abuse (R01 AI147490), and the University of San Diego Center for AIDS

Research (CFAR), a NIH funded program (P30 AI036214). Adriane Wynn acknowledges funding from

the National Institute on Alcohol Abuse and Alcoholism (K01 AA027733) and the National Institute on

Drug Abuse (T32 DA023356)."

3. Please include your tables as part of your main manuscript and remove the individual files. Please note that supplementary tables should remain uploaded as separate "supporting information" files.

5. We noticed you have some minor occurrence of overlapping text with the following previous publication, which needs to be addressed:

https://liverfoundation.org/wp-content/uploads/2020/01/Eliminate-Hepatitis-C-Initiative-Recommendations-to-Board-of-Supervisors-12.20.19.pdf

The text that needs to be addressed involves the following sections:

-Results section, paragraph 2, sentence 2

-Discussion section, paragraph 2, sentences 3-4

-Discussion sectino, paragraph 3, sentences 2-3

In your revision ensure you cite all your sources (including your own works), and quote or rephrase any duplicated text outside the methods section. Further consideration is dependent on these concerns being addressed.

Reviewers' comments:

Reviewer's Responses to Questions

**Comments to the Author**

1. Is the manuscript technically sound, and do the data support the conclusions?

Reviewer #1: Partly

Reviewer #2: No

2. Has the statistical analysis been performed appropriately and rigorously? 

Reviewer #1: Yes

Reviewer #2: I Don't Know

3. Have the authors made all data underlying the findings in their manuscript fully available?

Reviewer #1: Yes

Reviewer #2: No

4. Is the manuscript presented in an intelligible fashion and written in standard English?

Reviewer #1: Yes

Reviewer #2: No

5. Review Comments to the Author

Reviewer #1: The manuscript "Estimated hepatitis C prevalence and key population sizes in San Diego" is interesting and can help with strategies to combat hepatitis C in San Diego. However, the authors must adjust some points for the final publication.

First, the title must include the year 2018, as specified in the objective at the end of the introduction.

Major coments

Should authors inform the possible reasons (transfusion, drug use, ...) for HCV prevalence to be higher in the 55-74 age group, both in men and women in the general population? What is the average age of PWIDs and MSM with HCV? What are the epidemiological characteristics of the population incarcerated with HCV?

Authors should put in the conclusions which are the possible interventions and which are the priority groups for these. Are only people between 55-74 years old and PWID? How can the results help in making political decisions to better fight HCV infection?

Minor comments

In the summary, the authors cite "general adult populations and subpopulations" but must include what the subpopulations are, as they quote the acronym PWID in Findings, without citing previously. The reader may be confused if he does not know what the acronym means. Still in the Abstract and in the methodology, the authors must inform that only the results of the population in general had the stratification of sex and age, since the other groups did not have this information.

In the methodology, the authors need to better describe how information about HIV will be obtained in MSM.

Reviewer #2: General

Since hepatitis C infection mostly remains silent due to its asymptomatic nature, the infected persons remain unaware of their clinical status until cirrhosis, liver decompensation or HCC occurred. Increasing the number of HCV diagnosed patients, and subsequently linked them to appropriate care, is a crucial step toward achieving the WHO goal of HCV eradication. Global control of HCV infection becomes feasible but depends on the capacity of countries to identify infected people and to offer them treatment. This study aimed to estimate the HCV burden in San Diego County and to identify HCV-infected individuals who are currently unaware of their HCV status. The main question addressed by the authors is relevant and interesting. However, in my opinion important pittfalls are present.

Major concerns

-In the current study, the authors estimated HCV burden in San Diego County using only data on HCV seroprevalence (anti-HCV positivity). A positive anti-HCV test is indicative of exposure to HCV, whereas viremic infection (i.e. ongoing infection) as positive anti-HCV and HCV RNA, and is indicative of a chronic or acute HCV infection. In my opinion the lack of data on HCV-RNA is a strong limit for the purpose of this study. A previous study which estimated HCV burden in the San Francisco population synthesizing multiple data sources (triangulation approach) to produce a reliable baseline estimate of the number of people in San Francisco with anti-HCV antibodies (`seropositive') and active HCV infection (`viremic'). (Facente SN, Grebe E, Burk K, Morris MD, Murphy EL, Mirzazadeh A, et al. (2018) Estimated hepatitis C prevalence and key population sizes in San Francisco: A foundation for elimination. PLoS ONE 13(4): e0195575. https://doi.org/10.1371/ journal.pone.0195575

- Methods section is not clear, confuse and not easy to read. The triangulation approach is not described in the text but only cited in tha abstract “we triangulated data.,,,”. Therefore, it is not clear as this approach was conducted and the statistical analysis performed.

Minor comments:

- Title should be changed in "Estimated hepatitic c seroprevalence...."

- Abstract: The study aim is reported in the text as a finding; it could be replaced as “Our analysis was designated to estimate ….”.

- Methods pag 4: “Due to a lack of data on HCV in children (aged<18), this group was excluded from the analysis”. The authors declared to estimate the HCV burden in the adult population thus this sentence is not necessary.

- Methods pag 6: In the first paragraph, the authors cited” Additionally, it is acknowledged that the 1945-1965 birth cohort and general population may have also been infected by the use of injection drugs, thus the PWID population was distinguished as ”current PWID”, separate from those infected via contaminated needles, but no longer injecting.” , but the reference is missing.

6. PLOS authors have the option to publish the peer review history of their article (what does this mean?). If published, this will include your full peer review and any attached files.

Reviewer #1: **Yes: **Luiz Fernando Almeida Machado

Reviewer #2: No

---

## [Author Response · Author response to Decision Letter 0]

28 Apr 2021

Please see our cover letter for a thorough response to reviewers. 

Reviewers' comments:

Reviewer #1, comment #1: The manuscript "Estimated hepatitis C prevalence and key population sizes in San Diego" is interesting and can help with strategies to combat hepatitis C in San Diego. However, the authors must adjust some points for the final publication. First, the title must include the year 2018, as specified in the objective at the end of the introduction.

Author reply: Thank you, we have now revised the title to include 2018. 

Title: Estimated hepatitis C seroprevalence and key population sizes in San Diego in 2018

Reviewer #1, comment #2: Should authors inform the possible reasons (transfusion, drug use, ...) for HCV prevalence to be higher in the 55-74 age group, both in men and women in the general population? What is the average age of PWIDs and MSM with HCV? What are the epidemiological characteristics of the population incarcerated with HCV?

Author reply: We thank the reviewer for this important comment. We have now included an explanation for the increased HCV risk among the birth cohort and characteristics of other risk groups in the discussion.

Methods (page 5): According to the Centers for Disease Control and Prevention (CDC), the higher HCV seroprevalence among adults born from 1945-1965 corresponds to the high number of incident infections that occurred among young adults in the 1970s and 1980s due to the high frequency of injecting drug use during that period or receipt of contaminated blood transfusions(8).

Discussion (pages 11-12): Meeting the mortality goal requires treating individuals with more advanced liver disease; these individuals may be older without ongoing transmission risk, such as those in the 1945-1965 birth cohort. Whereas, reducing incidence requires treatment and prevention interventions for those with ongoing risk such as PWID; these individuals may be younger with less advanced disease. Largely due to the expanding opioid epidemic,(2) acute HCV diagnoses are increasing in the U.S., particularly among younger PWID who are between 18 and 35 years old.(3, 4).

Reviewer #1, comment #2: Authors should put in the conclusions which are the possible interventions and which are the priority groups for these. Are only people between 55-74 years old and PWID? How can the results help in making political decisions to better fight HCV infection?

Author reply: We thank the reviewer for this important point and agree that discussion of the relevant interventions and implications of our findings was insufficiently described. We have now added a paragraph to the discussion section on interventions.

Discussion (page 11) Importantly, our study also found that nearly one-fifth of infections were among the general population outside of the 1945-1965 birth cohort, indicating the importance of surveillance, prevention, testing, and linkages to care in this group as well. In 2020, the CDC expanded HCV screening guidelines to recommend a one-time screen for all adults aged 18 years and older, in addition to risk-based screening.(20) Although the COVID-19 pandemic may have interrupted HCV screening efforts,(21) universal screening of adults in San Diego could enable the detection of a substantial fraction of HCV infections. Following diagnosis, short-duration direct-acting antivirals (DAAs) are highly effective at curing >90% of individuals,(3), preventing HCV-related mortality and could also prevent transmission.(22)

Discussion (pages 11-12) Largely due to the expanding opioid epidemic,(2) acute HCV diagnoses are increasing in the U.S., particularly among younger PWID who are between 18 and 35 years old.(3, 4). The risks associated with injecting drug use are estimated to account for 77% of ongoing HCV transmission in North America.(23) Harm reduction interventions such as opiate substitution therapy (OST) and needle/syringe programs (NSPs) have been found to be both effective and cost-effective to prevent HCV acquisition.(24) Thus, scale-up of combination harm reduction and treatment services are required to achieve incidence reduction goals.(22, 25)

Minor comments

Reviewer #1, comment #3: In the summary, the authors cite "general adult populations and subpopulations" but must include what the subpopulations are, as they quote the acronym PWID in Findings, without citing previously. The reader may be confused if he does not know what the acronym means. Still in the Abstract and in the methodology, the authors must inform that only the results of the population in general had the stratification of sex and age, since the other groups did not have this information.

Author reply: We apologize for the lack of clarity in the abstract. Based on your comments, we have revised the abstract to include detail of the subpopulations and define the PWID acronyms. We have also edited the abstract and methods to clarify that only the general population was stratified by age and sex. 

Abstract (page 2) Burden estimates were stratified by subgroup (people who inject drugs in the community [PWID], men who have sex with men in the community [MSM], general population in the community [stratified by age and sex], and incarcerated individuals). 

Methods (page 4) The mutually exclusive groups examined were: people who inject drugs (PWID) in the community, defined as those who have injected in the past 12 months; men who have sex with men (MSM) in the community; the general population in the community, excluding the aforementioned groups (general population stratified by age and sex); and people who are incarcerated in California with San Diego as their county of commitment.

Reviewer #1, comment #4: In the methodology, the authors need to better describe how information about HIV will be obtained in MSM.

Author reply: We apologize for the confusion. Data on the HIV prevalence among MSM in San Diego was obtained from the 2017 National HIV Behavioral Surveillance Survey among MSM in San Diego. We now add additional specificity about this in the methods:

 Methods (page 8) As such, an estimate of the number of MSM with HIV was first generated by multiplying the aforementioned MSM population size estimate by the weighted HIV prevalence observed among MSM from the 2017 San Diego NHBS MSM survey data (20%; 95% CI: 11.1-28.9%, sampled from a beta distribution). 

Reviewer #2: 

General

Since hepatitis C infection mostly remains silent due to its asymptomatic nature, the infected persons remain unaware of their clinical status until cirrhosis, liver decompensation or HCC occurred. Increasing the number of HCV diagnosed patients, and subsequently linked them to appropriate care, is a crucial step toward achieving the WHO goal of HCV eradication. Global control of HCV infection becomes feasible but depends on the capacity of countries to identify infected people and to offer them treatment. This study aimed to estimate the HCV burden in San Diego County and to identify HCV-infected individuals who are currently unaware of their HCV status. The main question addressed by the authors is relevant and interesting. However, in my opinion important pittfalls are present.

Major concerns

Reviewer #1, comment 1: In the current study, the authors estimated HCV burden in San Diego County using only data on HCV seroprevalence (anti-HCV positivity). A positive anti-HCV test is indicative of exposure to HCV, whereas viremic infection (i.e. ongoing infection) as positive anti-HCV and HCV RNA, and is indicative of a chronic or acute HCV infection. In my opinion the lack of data on HCV-RNA is a strong limit for the purpose of this study. A previous study which estimated HCV burden in the San Francisco population synthesizing multiple data sources (triangulation approach) to produce a reliable baseline estimate of the number of people in San Francisco with anti-HCV antibodies (`seropositive') and active HCV infection (`viremic'). (Facente SN, Grebe E, Burk K, Morris MD, Murphy EL, Mirzazadeh A, et al. (2018) Estimated hepatitis C prevalence and key population sizes in San Francisco: A foundation for elimination. PLoS ONE 13(4): e0195575. https://doi.org/10.1371/ journal.pone.0195575

Author reply: We thank the reviewer for raising this important point and fully agree that the lack of data related to viremic infections in San Diego County is an important limitation to our study. We agree that a similar study in San Francisco was able to generate an estimate of both the seroprevalence and viremic infection burden. Importantly, the San Francisco study authors (Facente et al.) were able to generate viremic infection estimates because of their partnership with the Blood Systems Research Institute, which conducted RNA testing among blood donors in the adult general population. Unfortunately, no similar data are available within San Diego County, as the existing blood banks only test HCV antibodies. Thus, we were unable to generate an estimate of viremic prevalence, and agree this is an important limitation as ongoing monitoring of elimination progress would ideally track changes in viremic infections. We have made a number of edits to clarify and address this important limitation. First, we have updated our title to reflect that the estimate is for seroprevalence. Further, we added additional details in the discussion around this important limitation:

Discussion (page 12): Second, we were unable to determine viremic chronic infection burden as RNA data were not available from blood donors in San Diego County. An estimated one-quarter of individuals spontaneously clear infection and do not progress to chronic infection, with slightly lower clearance rates among individuals with HIV.(26, 27) Information on chronic (active) infection is essential for determining HCV treatment need in San Diego County and for monitoring progress towards elimination as individuals are cured. Information on chronic (active) infection is therefore essential for determining HCV treatment need in San Diego County and for monitoring progress towards elimination as individuals are cured. 

Reviewer #1, comment 2: Methods section is not clear, confuse and not easy to read. The triangulation approach is not described in the text but only cited in tha abstract “we triangulated data.,,,”. Therefore, it is not clear as this approach was conducted and the statistical analysis performed.

Author reply: We apologize for the lack of clarity in the methods section. To address this concern, we have now added a paragraph to the beginning of the methods section which outlines our approach to aid in clarity. Further, we have removed the confusing term, “triangulation,” as we agree this was not the correct term to describe our methods. We hope our edited text below has aided in clarifying our approach. 

Abstract/Methdods (page 2): We synthesized data from the American Community Survey, Centers for Disease Control and Prevention, California Department of Public Health, Public Health Branch of California Correctional Health Care Services, San Diego Blood Bank, and published literature.

Methods (page 5): First, we estimated the population sizes of the above-mentioned groups using data from the American Community Survey (ACS), San Diego National HIV Behavioral Survey (NHBS), the Public Health Branch of California Correctional Health Care Services, and published literature. Next, we estimated HCV seroprevalence for each group using data from the San Diego Blood Bank, the Study of Tuberculosis, AIDS, and Hepatitis C Risk (STAHR II), the NHBS, the Public Health Branch of California Correctional Health Care Services, and published literature. Finally, we multiplied the prevalence by the population size for each group to obtain estimates of the number of individuals with current or past HCV in San Diego County.

Reviewer #2, comment 3: Title should be changed in "Estimated hepatitic c seroprevalence...."

Author reply: We have updated the title as you suggest.

Reviewer #2, comment 4: The study aim is reported in the text as a finding; it could be replaced as “Our analysis was designated to estimate ….”.

Author reply: We have revised this sentence: 

Abstract (page 2): “Our analysis was designed to estimate the HCV burden in San Diego County in 2018.”

Reviewer #2, comment 5: Methods pag 4: “Due to a lack of data on HCV in children (aged<18), this group was excluded from the analysis”. The authors declared to estimate the HCV burden in the adult population thus this sentence is not necessary.

Author reply: We have removed this sentence as requested.

Reviewer #2, comment 6: Methods pag 6: In the first paragraph, the authors cited” Additionally, it is acknowledged that the 1945-1965 birth cohort and general population may have also been infected by the use of injection drugs, thus the PWID population was distinguished as ”current PWID”, separate from those infected via contaminated needles, but no longer injecting.” , but the reference is missing.

Author reply: We agree that this sentence was confusing. We have now deleted this sentence and instead inserted a new sentence at the beginning of the PWID population size paragraph clarifying that we define PWID as those with recent (past year) injecting drug use. Therefore, by definition any past PWID would be captured in our general population estimate. 

Methods (page 5) We define the PWID population as individuals with recent (past year) injecting drug use; therefore, individuals with past injecting drug use risk would be classified as part of the general population in our estimates).

References

1. Centers for Disease Control and Prevention. Viral Hepatitis Surveillance Report 2018 - Hepatitis C. 2020.

2. Schwetz TA, Calder T, Rosenthal E, Kattakuzhy S, Fauci AS. Opioids and Infectious Diseases: A Converging Public Health Crisis. J Infect Dis. 2019.

3. Centers for Disease Control and Prevention. National Notifiable Diseases Surveillance System. Accessed from: https://wwwcdcgov/mmwr/mmwr_nd/indexhtml. 2019.

4. Abara WE, Trujillo L, Broz D, Finlayson T, Teshale E, Paz-Bailey G, et al. Age-Related Differences in Past or Present Hepatitis C Virus Infection Among People Who Inject Drugs: National Human Immunodeficiency Virus Behavioral Surveillance, 8 US Cities, 2015. The Journal of Infectious Diseases. 2019;220(3):377-85.

5. Varan AK, Mercer DW, Stein MS, Spaulding AC. Hepatitis C seroprevalence among prison inmates since 2001: still high but declining. Public health reports (Washington, DC : 1974). 2014;129(2):187-95.

6. Spaulding AC, Anderson EJ, Khan MA, Taborda-Vidarte CA, Phillips JA. HIV and HCV in U.S. Prisons and Jails: The Correctional Facility as a Bellwether Over Time for the Community's Infections. AIDS reviews. 2017;19(3):134-47.

---

## [Editor Report · Decision Letter 1]

30 Apr 2021

The Estimated hepatitis C seroprevalence and key population sizes in San Diego in 2018

PONE-D-20-38492R1

Dear Dr. Wynn,

We’re pleased to inform you that your manuscript has been judged scientifically suitable for publication and will be formally accepted for publication once it meets all outstanding technical requirements.

Kind regards,

Yury E Khudyakov, PhD

Academic Editor

PLOS ONE
---

## [Editor Report · Acceptance letter]

31 May 2021

PONE-D-20-38492R1 

The Estimated hepatitis C seroprevalence and key population sizes in San Diego in 2018 

Dear Dr. Wynn:

I'm pleased to inform you that your manuscript has been deemed suitable for publication in PLOS ONE. Congratulations! Your manuscript is now with our production department. 

Kind regards, 

on behalf of

Dr. Yury E Khudyakov 

Academic Editor

PLOS ONE